# ETGL-DDPG: A Deep Deterministic Policy Gradient Algorithm for Sparse Reward Continuous Control

## Abstract

We consider deep deterministic policy gradient (DDPG) in the context of reinforcement learning with sparse rewards. To enhance exploration, we introduce a search procedure, $\epsilon t$-*greedy*, which generates exploratory options for exploring less-visited states. We prove that search using $\epsilon t$-greedy has polynomial sample complexity under mild MDP assumptions. To more efficiently use the information provided by rewarded transitions, we develop a new dual experience replay buffer framework, *GDRB*, and implement *longest n-step returns*. The resulting algorithm, *ETGL-DDPG*, integrates all three techniques: $\boldsymbol{\epsilon t}$-greedy, **G**DRB, and **L**ongest $n$-step, into DDPG. We evaluate ETGL-DDPG on standard benchmarks and demonstrate that it outperforms DDPG, as well as other state-of-the-art methods, across all tested sparse-reward continuous environments. Ablation studies further highlight how each strategy individually enhances the performance of DDPG in this setting.

## 1 Introduction

Deep deterministic policy gradient (DDPG) (Lillicrap et al., 2015) is one of the representative algorithms for reinforcement learning (RL) (Sutton & Barto, 2018), alongside other prominent approaches (Haarnoja et al., 2018; Fujimoto et al., 2018; Andrychowicz et al., 2017). The method has been extensively used for continuous control environments with dense reward signals (Duan et al., 2016). However, its performance degrades significantly when the reward signals are sparse and are only observed upon reaching the goal (Matheron et al., 2019).

In sparse-reward environments where success depends on reaching a goal state, DDPG's deficiency can be explained from three perspectives. The first one is its lack of *directional exploration*. Like other off-policy RL algorithms, DDPG employs a *behavior policy* for exploring the environment. The standard choices are either an $\epsilon$-greedy behavior policy that samples a random action with probability $\epsilon$ (e.g., $0.1$), or the main policy with artificial noise. As argued in (Dabney et al., 2020), these one-step *noise augmented greedy* strategies are ineffective for exploring large sparse-reward state spaces due to the lack of temporal abstraction. To improve $\epsilon$-greedy, Dabney et al. (2020) propose a temporally extended $\epsilon z$-greedy policy that expands exploration into multiple steps, controlled by a distribution $z$. $\epsilon z$-greedy represents an advancement from the option framework for reinforcement learning (Sutton et al., 1999). Theoretically, an option $O$ is defined as a tuple $O = \langle I, \pi, \beta \rangle$, where $I$ is the set of states where an option can begin, $\pi$ is the option policy that determines which actions to take while executing the option, and $\beta$ is the termination condition. In $\epsilon z$-greedy, each option repeats a primitive action for a specific number of time steps which is sampled from a distribution $z$ (e.g., a uniform distribution). The option can begin at any state with probability $\epsilon$ and terminates whenever their length reaches a limit that is decided by $z$. While $\epsilon z$-greedy improves over $\epsilon$-greedy, it is also *directionless*: for exploratory action, the agent does not use any information from its experience for more informed exploration.

The second drawback of DDPG is its uniform treatment of zero and non-zero rewards in the replay buffer. For most off-policy RL algorithms, a replay buffer is used to store and sample transitions of the agent's interactions with the environment. By default, DDPG uses a uniform sampling strategy that assigns an equal probability of being chosen to all transitions in the buffer. In sparse-

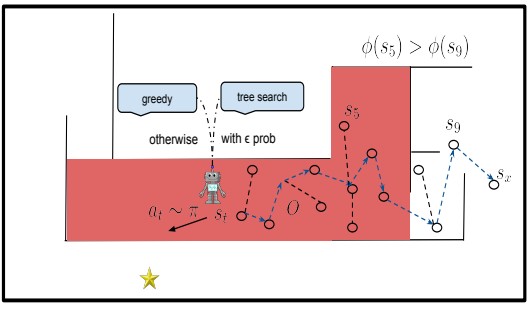

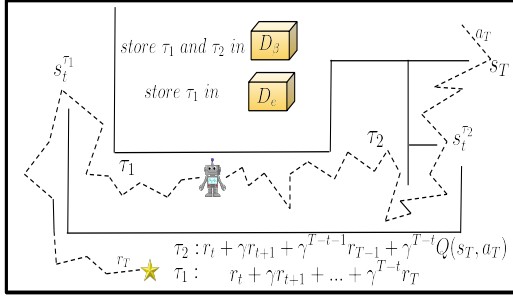

(a) $\epsilon t$-greedy: greedy or tree search

(b) GDRB and the longest n-step return

Figure 1: (a): $\epsilon t$-greedy exploration strategy. The agent creates a tree from the current state $s_t$ with $\epsilon$ probability. Otherwise, it uses its policy to determine the next action $a_t \sim \pi$. The tree uses a hash function $\phi$ to estimate the visit counts to states. If the newly added node $s_x$ to the tree is located in an unvisited area $n(\phi(s_x)) = 0$, the path from the root to that node is returned as option $O$. The tree helps in avoiding obstacles, discovering unexplored areas, and staying away from highly-visited regions (middle red area). (b): GDRB and the longest n-step return for Q-value updates. $\tau_1$ reaches the goal (a successful episode), and $\tau_2$ is truncated by time limit (an unsuccessful episode). The first buffer $D_\beta$ stores both trajectories but $D_e$ only stores successful trajectories. The target Q-value for state $s_t$ is shown for both trajectories below the figure. In successful episodes, the target Q-value is the episode return. $s_T$ represents the last state in each episode, which is the goal state indicated by a star in $\tau_1$.

reward environments, uniform sampling therefore rarely chooses rewarded transitions. In general, RL algorithms can be improved by prioritizing transitions based on the associated rewards or TD error (Schaul et al., 2015). For problems with well-defined goals, a replay buffer can be further enhanced to exploit the strong correlation of rewards and goals. The third weakness of DDPG is its slow information propagation when updating its learning policy. Since only the last transition in a successful episode (i.e., goal reached) gets rewarded, in standard DDPG, the agent must achieve the goal many times to make sure that the reward is eventually propagated backward to early states. It is known that one way to achieve this is to provide intermediate rewards with reward shaping methods (Laud, 2004). However, effective reward shaping is usually problem-specific and does not generalize to a wide range of tasks.

In this paper, we enhance DDPG (Lillicrap et al., 2015) to address all three aforementioned problems. Our first contribution is $\epsilon t$-*greedy*, a new temporally version of $\epsilon$-greedy that utilizes a light-weight search procedure, similar to Laud (2004), to enable more directional exploration based on the agent's previous experience data. We show that similar to $\epsilon z$-greedy, $\epsilon t$-greedy has polynomial sample complexity in related parameters of the MDP. Our second contribution is a new *goal-conditioned dual replay buffer* (GDRB), that uses two replay buffers along with an adaptive sampling strategy to differentiate goal-reached and goal-not-reached experience data. These two buffers differ in retention policy, size, and the transitions they store. Our third enhancement is to replace the one-step update in DDPG with the longest $n$-step return for all transitions in an episode. Figure 1 illustrates the innovations of ETGL-DDPG. In Section 4, we evaluate the performance of ETGL-DDPG through extensive experiments on 2D and 3D continuous control benchmarks. We show that each of the three strategies individually improves the performance of DDPG. Furthermore, ETGL-DDPG outperforms current state-of-the-art methods across all tested environments.

## 2 BACKGROUND

We consider a Markov decision process (MDP) defined by the tuple $(S, A, \mathcal{T}, r, \gamma, \rho)$. $S$ is the set of states, $A$ is the set of actions, $\mathcal{T}(s'|s, a)$ is the transition distribution, $r : S \times A \times S \to \mathbb{R}$ is the reward function, $\gamma \in [0, 1]$ is the discount factor, and $\rho(s_0, s_g)$ is the distribution from which initial and goal states are sampled for each episode. Every episode starts with sampling a new pair of initial and goal states. At each time-step $t$, the agent chooses an action using its policy and considering the current state and the goal state $a_t = \pi(s_t, s_g)$ resulting in reward $r_t = (s_t, a_t, s_g)$. The next state is sampled

from $\mathcal{T}(.|s_t, a_t)$. The episode ends when either the goal state or the maximum number of steps $T$ is reached. The return is the discounted sum of future rewards $R_t = \sum_{i=t}^{T} \gamma^{i-t} r_i$. The Q-function and value function associated with the agent's policy are defined as $Q^\pi(s_t, a_t, s_g) = \mathbb{E}[R_t|s_t, a_t, s_g]$ and $V^\pi(s_t, s_g) = max_a Q^\pi(s_t, a_t, s_g)$. The agent's objective is to learn an optimal policy $\pi^*$ that maximizes the expected return $\mathbb{E}_{s_0}[R_0|s_0, s_g]$.

## 2.1 DEEP DETERMINISTIC POLICY GRADIENT (DDPG)

To ease presentation, we adopt our notation with explicit reference to the goal state for both the critic and the actor networks in DDPG. DDPG maintains an actor $\mu(s, s_g)$ and a critic $Q(s, a, s_g)$. The agent explores the environment through a stochastic policy $a \sim \mu(s, s_g) + w$, where $w$ is a noise sampled from a normal distribution or an Ornstein-Uhlenbeck process (Uhlenbeck & Ornstein, 1930). To update both actor and critic, transition tuples are sampled from a replay buffer to perform a mini-batch gradient descent. The critic is updated by a loss $L$,

$$L = \mathbb{E}[Q(s_t, a_t, s_g) - y_t]^2 \tag{1}$$

where $y_t = r_t + \gamma Q'(s_{t+1}, \mu'(s_{t+1}, s_g), s_g)$. $Q'$ and $\mu'$ are the target critic and actor, respectively; their weights are soft-updated to the current weights of the main critic and actor, respectively. The actor is updated by the deterministic policy gradient algorithm (Silver et al., 2014) to maximize the estimated Q-values of the critic using loss $-\mathbb{E}_s[Q(s, \mu(s, s_g), s_g)]$.

## 2.2 LOCALITY-SENSITIVE HASHING

Our approach discretizes the state space with a hash function $\phi : \mathbb{S} \rightarrow \mathbb{M}$, that maps states to buckets in $\mathbb{M}$. When we encounter a state $s$, we increment the visit count for $\phi(s)$. We use $n(\phi(s))$ as the visit counts of all states that map to the same bucket $\phi(s)$. Clearly, the *granularity* of the discretization significantly impacts our exploration method. The goal for the granularity is that "distant" states are in separate buckets while "similar" states are grouped into one.

We use Locality-Sensitive Hashing (LSH) as our hashing function, a popular class of hash functions for querying nearest neighbors based on a similarity metric (Bloom, 1970). SimHash (Charikar, 2002) is a computationally efficient LSH method that calculates similarity based on angular distance. SimHash retrieves a binary code of state $s \in S$ as

$$\phi(s) = sgn(Af(s)) \in \{-1, 1\}^k, \tag{2}$$

where $f : S \rightarrow \mathbb{R}^D$ is a preprocessing function and $A$ is a $k \times D$ matrix with i.i.d. entries drawn from a standard Gaussian distribution $\mathcal{N}(0, 1)$. The parameter $k$ determines the granularity of the hash: larger values result in fewer collisions, thereby enhancing the ability to distinguish between different states.

# 3 THE ETGL-DDPG METHOD

In this section, we describe three strategies in ETGL-DDPG for improving DDPG in sparse-reward tasks. The full pseudocode for ETGL-DDPG is presented in Supplementary Algorithm 3.

## 3.1 $\epsilon t$-GREEDY: EXPLORATION WITH SEARCH

In principle, exploration should be highest at the beginning of training, as discovering rewarded transitions during early steps is essential for escaping local optima (Matheron et al., 2019). Motivated by the success of the fast exploration algorithms RRT (LaValle, 1998) and $\epsilon z$-greedy (Dabney et al., 2020), we introduce $\epsilon t$-*greedy*, which combines $\epsilon$-greedy with a *tree search* procedure. Like $\epsilon$-greedy, $\epsilon t$-greedy selects a greedy action with probability $1 - \epsilon$, and an exploratory action with probability $\epsilon$. However, instead of exploring uniformly at random, the exploratory action in $\epsilon t$-greedy is the first step of an *option* generated via a search with time budget $N$.

To execute the search process, the agent requires access to the environment's transition function $\mathcal{T}$ of the corresponding MDP. This is used to generate new nodes within the search tree. However, since our exploration strategy is built on DDPG, the model-free algorithm, the transition function $\mathcal{T}$ is not known. Instead, the agent utilizes its replay buffer to advance the search. We briefly discuss the impact of having access to $\mathcal{T}$ on the exploration process in Supplementary Material A.2. We also assume that the agent has a SimHash function $\phi$, which discretizes the large continuous environment. For each state $s$, $n(\phi(s))$ serves as an estimate of the number of visits to a neighbourhood of $s$ throughout the entire learning process.

The replay buffer contains transitions observed during training. It can be used as a transition model for observed transitions and an approximate one for transitions similar to those already seen. For simplicity, we identify each bucket with its hash code $\phi(s)$. We use a buffer $B_M$ which stores observed transitions based on the hash of their states $\phi(s)$. If the agent makes a transition $(s_t, a_t, r_t, s_{t+1})$ in the environment, the transition is stored in bucket $b = \phi(s_t)$. All transitions are assigned to their buckets upon being added to the replay buffer. As training may take a long time, we limit the number of transitions in each bucket, and randomly replace one of the old transitions in a full bucket with the new transition.

The function `next_state_from_replay_buffer` in Algorithm 1 shows how new nodes can be added to the search: assuming we are at node $s_x$, we randomly select a transition $(s', a, r, s'')$ in bucket $\phi(s_x)$ and create a new child $s_{x'}$ for $s_x$ by using following approximation:

$$\mathcal{T}(s_x, a) \approx \mathcal{T}(s', a) \tag{3}$$

Algorithm 1 explains how the search generates an exploratory option. Initially, at state $s$, we create a list of frontier nodes consisting of only the root node $s$. If bucket of state $s$ in $B_M$ is empty: $b_{\phi(s)} = \varnothing$, there is no transition to approximate $\mathcal{T}(s, a)$. In this case, $\epsilon t$-greedy as in $\epsilon$-greedy generates a random action at $s$. Otherwise, when $b_{\phi(s)} \neq \varnothing$, $\epsilon t$-greedy conducts a tree search iteratively, with a maximum of $N$ iterations. At each iteration, a node $s_x$ is sampled uniformly from the frontier nodes, and a *child* for $s_x$, noted as $s_{x'}$, is generated using `next_state_from_replay_buffer` function. If $n(\phi(s_{x'})) = 0$, we terminate and return the action sequence from the root to $s_{x'}$; otherwise, we repeat this process until we have added $N$ nodes to the tree. We then return the action sequence from the root to a least-visited node $s_{min}$:

$$s_{min} = \min_{s \in frontier\ nodes} n(\phi(s)) \tag{4}$$

To justify this exploration method, we adopt the conditions outlined in Liu & Brunskill (2018) to validate the sample efficiency of $\epsilon t$-greedy. We begin by introducing the relevant terms and then present the main theorem. Detailed definitions and proofs are provided in Appendix A.1. The key idea is to define a measure that captures the concept of visiting all state-action pairs, as outlined in Definition 1.

**Definition 1** (**Covering Length**). *The covering length (Even-Dar & Mansour, 2004) represents the minimum number of steps an agent must take in an MDP, starting from any state-action pair $(s, a) \in \mathcal{S} \times \mathcal{A}$, to visit all state-action pairs at least once with a probability of at least 0.5.*

Our objective is to find a near-optimal policy, as defined in Definition 2.

**Definition 2** ($\epsilon$-**optimal Policy**). *A policy $\pi$ is called $\delta$-optimal if it satisfies $V^{\pi^*}(s) - V^{\pi}(s) \leq \epsilon$, for all $s \in S$, where $\epsilon > 0$.*

Next, we define the concept of sample efficiency, which is captured through the notion of polynomial sample complexity in Definition 3.

**Definition 3** (**PAC-MDP Algorithm**). *Given a state space $\mathcal{S}$, action space $\mathcal{A}$, suboptimality error $\epsilon > 0$ (from Definition 2) and $0 < \delta < 1$, an algorithm $\mathcal{A}$ is called PAC-MDP (Kakade, 2003), if the number of time steps required to find a $\epsilon$-optimal policy is less than some polynomial in the relevant quantities $(|\mathcal{S}|, |\mathcal{A}|, \frac{1}{\epsilon}, \frac{1}{1-\gamma}, \frac{1}{\delta})$ with probability at least $1 - \delta$.*

For simplicity, when we say an algorithm $\mathcal{A}$ has polynomial sample complexity, we imply that $\mathcal{A}$ is PAC-MDP. The work by Liu & Brunskill (2018) establishes polynomial sample complexity for

---

**Algorithm 1** Generating exploratory option with tree search

---

1: **function generate_option**(state s, hash function $\phi$, budget N)
2:     frontier_nodes $\leftarrow \{\}$
3:     Initialize root using $s$: $root \leftarrow TreeNode(s)$
4:     frontier_nodes $\leftarrow$ frontier_nodes $\cup \{root\}$;
5:     $s_{\min} \leftarrow$ root
6:     $i \leftarrow 0$
7:     **while** $i < N$ **do**
8:         $s_x \sim UniformRandom$(frontier_nodes)
9:         $s_{x'} =$ **next_state_from_buffer**($s_x$)
10:        **if** $n(\phi(s_{x'}))=0$ **then**
11:            Extract option $o$ by actions $root$ to $s_{x'}$
12:            **return** $o$
13:        **end if**
14:        **if** $n(\phi(s_{x'})) < n(\phi(s_{\min}))$ **then**
15:            $s_{\min}=s_{x'}$
16:        **end if**
17:        $i \leftarrow i + 1$
18:    **end while**
19:    Extract option $o$ by actions $root$ to $s_{\min}$
20:    **return** $o$
21: **end function**
22:
23: **function next_state_from_buffer**($s_x$, frontier_nodes)
24:     $(s', a, r, s'') \sim UnifromRandom(\phi(s_x))$
25:     $s_{x'} \leftarrow s''$
26:     $s_x$.add_child($s_{x'}$)
27:     frontier_nodes $\leftarrow$ frontier_nodes $\cup \{s_{x'}\}$
28:     **return** $s_{x'}$
29: **end function**

---

a uniformly random exploration by bounding the covering length defined in Definition 1. Using this, and considering a limited tree budget $N$, we show that $\epsilon t$-greedy is PAC-MDP. Let's denote the search tree by $\mathcal{X}$, and the distribution over the generated options in $\mathcal{X}$ as $\mathcal{P}_\omega$. The following Theorem provides a lower bound on option sampling in tree $\mathcal{X}$ under certain condition.

**Theorem 1 (Worst-Case Sampling).** *Given a tree $\mathcal{X}$ with $N$ nodes ($s_1$ to $s_N$), for any $\omega \in \Omega_{\mathcal{X}}$, the sampling probability satisfies:*

$$\mathcal{P}_{\mathcal{X}}[\omega] \geq \frac{1}{N!(\max_{i \in [N]} |\phi(s_i)|)^{N-1}} \geq \frac{1}{\Theta(|\mathcal{S}||\mathcal{A}|)} \tag{5}$$

*, **if** $N \leq \frac{\log(|\mathcal{S}||\mathcal{A}|)}{\log\log(|\mathcal{S}||\mathcal{A}|)}$. Here, $\mathcal{S}$ and $\mathcal{A}$ represent the state space and action space, respectively.*

To prove Theorem 1, we examine the construction of the "hardest option", $\hat{\omega} \in \Omega_{\mathcal{X}}$, which has the lowest sampling probability in the tree $\mathcal{X}$. Since $\mathcal{P}_{\mathcal{X}}$ is an unknown distribution, we cannot directly exploit it. Instead, we construct a worst-case scenario to approximate the minimum option sampling probability. Now, we present the following Theorem on the sample complexity of $\epsilon t$-greedy.

**Theorem 2 ($\epsilon t$-greedy Sample Efficiency).** *Given a state space $\mathcal{S}$, action space $\mathcal{A}$, and a set of options $\Omega_{\mathcal{X}}$ generated by $\epsilon t$-greedy for each tree $\mathcal{X}$, if $\mathcal{P}_{\mathcal{X}}[\omega] \geq \frac{1}{\Theta(|\mathcal{S}||\mathcal{A}|)}$, $\epsilon t$-greedy achieves polynomial sample complexity or i.e. is PAC-MDP.*

Theorem 1 asserts that the sampling bound condition from Theorem 2 is satisfied when $N \leq \frac{\log(|\mathcal{S}||\mathcal{A}|)}{\log\log(|\mathcal{S}||\mathcal{A}|)}$. Theorem 2 establishes the necessary lower bound on the sampling probability of an option $\omega \in \Omega_{\mathcal{X}}$ for any given exploration tree $\mathcal{X}$, ensuring that the $\epsilon t$-greedy strategy is PAC-MDP under this criterion.

## 3.2 GDRB: Goal-conditioned Dual Replay Buffer

The experience replay buffer is an indispensable part of deep off-policy RL algorithms. It is common to use only one buffer to store all transitions and use FIFO as the retention policy, with the most

recent data replacing the oldest data (Mnih et al., 2013). As an alternative, in the reservoir sampling (Vitter, 1985) retention policy, each transition in the buffer has an equal chance of being overwritten. This maintains coverage of some older data over training. *RS-DRB* (Zhang et al., 2019) uses two replay buffers, one for exploitation and the other for exploration. The transitions made by the agent's policy are stored in the exploitation buffer, and the random exploratory transitions are stored in the exploration buffer. For the retention policy, the exploration buffer uses reservoir sampling, while the exploitation buffer uses FIFO.

Inspired by this dual replay buffer framework, we propose a *Goal-conditioned Double Replay Buffer (GDRB)*. The first buffer $D_\beta$ stores all transitions during training, and the second buffer $D_e$ stores the transitions that belong to successful episodes (i.e., goal reached). $D_\beta$ uses reservoir sampling, and $D_e$ uses FIFO. Since $D_\beta$ needs to cover transitions from the entire training process, it is larger than $D_e$. We balance the number of samples taken from the two buffers with an adaptive sampling ratio. Specifically, in a training process of $E$ episodes, at current episode $i$, the sampling ratios $\tau_e$ and $\tau_\beta$ for $D_e$ and $D_\beta$ are set as follows: $\tau_e = \frac{i}{E}, \tau_\beta = 1 - \tau_e$. To select $C$ mini-batches, $\max(\lfloor \tau_\beta * C \rfloor, 1)$ mini-batches are chosen from $D_\beta$ and the rest from $D_e$. Later stages of training still sample from $D_\beta$ to not forget previously acquired knowledge, as we assume the policy is more likely to reach the goal as the training progresses. In case that $D_e$ is empty, since there are no successful episodes yet, we draw all mini-batches from $D_\beta$.

### 3.3 USING LONGEST $n$-STEP RETURN

In standard DDPG, $Q$-values are updated using one-step TD. In goal-reaching tasks with sparse rewards, only one rewarded transition per successful episode is added to the replay buffer. The agent needs rewards provided by these transitions to update its policy toward reaching the goal. With few rewarded transitions, the agent should exploit a successful path to the goal many times so the reward is propagated backward quickly. Multi-step updates can accelerate this process by looking ahead several steps, resulting in more rewarded transitions in the replay buffer (Meng et al., 2021; Hessel et al., 2018). For example, Meng et al. (2021) utilize $n$-step updates in DDPG with $n$ ranging from 1 to 8. In our design, to share the reward from the last step of a successful episode for all transitions in the episode, we use *longest $n$-step return* (Mnih et al., 2016), shown in Equation 6.

$$
Q(s_t, a_t) = \begin{cases} \sum_{k=0}^{T-t} \gamma^k r_{t+k}, & s_T \text{ is a goal state} \\ \sum_{k=0}^{T-t-1} \gamma^k r_{t+k} + \gamma^{T-t} Q(s_T, a_T), & \text{otherwise} \end{cases} \tag{6}
$$

Here, $s_T$ is the last state in the episode. Using the longest n-step return for each transition from a successful episode, the reward is immediately propagated to all $Q$-value updates. In unsuccessful episodes, using the longest $n$-step return reduces the overestimation bias in Q-values (Thrun & Schwartz, 1993). Meng et al. (2021) empirically show that using multi-step updates can improve the performance of DDPG on robotic tasks mostly by reducing overestimation bias — they demonstrate that the larger the number of steps, the lower the estimated target Q-value and overestimation bias.

## 4 EXPERIMENTS

In this section, we show the details of how ETGL-DDPG improves DDPG for sparse-reward tasks using its three strategies. We use experiments to answer the following questions: 1) Can ETGL-DDPG outperform state-of-the-art methods in goal-reaching tasks with sparse rewards? 2) How does each of these three innovations impact the performance of DDPG? 3) Can $\epsilon t$-greedy explore more efficiently than $\epsilon z$-greedy and other common exploration strategies?

We consider two types of tasks: *navigation* and *manipulation*. We use three sparse-reward continuous environments for navigation. The first environment is a 2D maze called *Wall-maze* (Trott et al., 2019), where a reward of -1 is given at each step, and a reward of 10 is given if the goal is reached. The start and goal states for each episode are randomly selected from the blue and green regions, respectively, as shown in Figure 2a. The agent's action (dx,dy) determines the amount of movement along both axes. The environment contains a gradient cliff feature (Lehman et al., 2018), where the fastest way to reach the goal results in a deadlock close to the goal. Our second and third 3D envi-

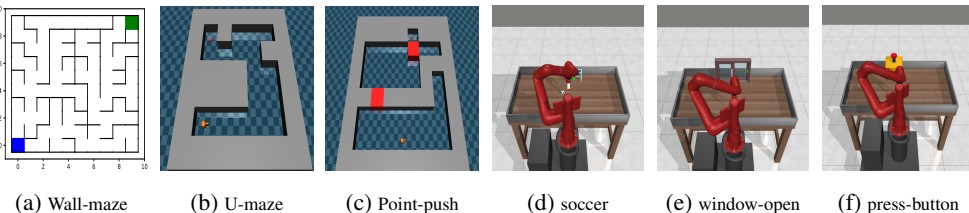

(a) Wall-maze    (b) U-maze    (c) Point-push    (d) soccer    (e) window-open    (f) press-button

Figure 2: The environments used in our experiments.

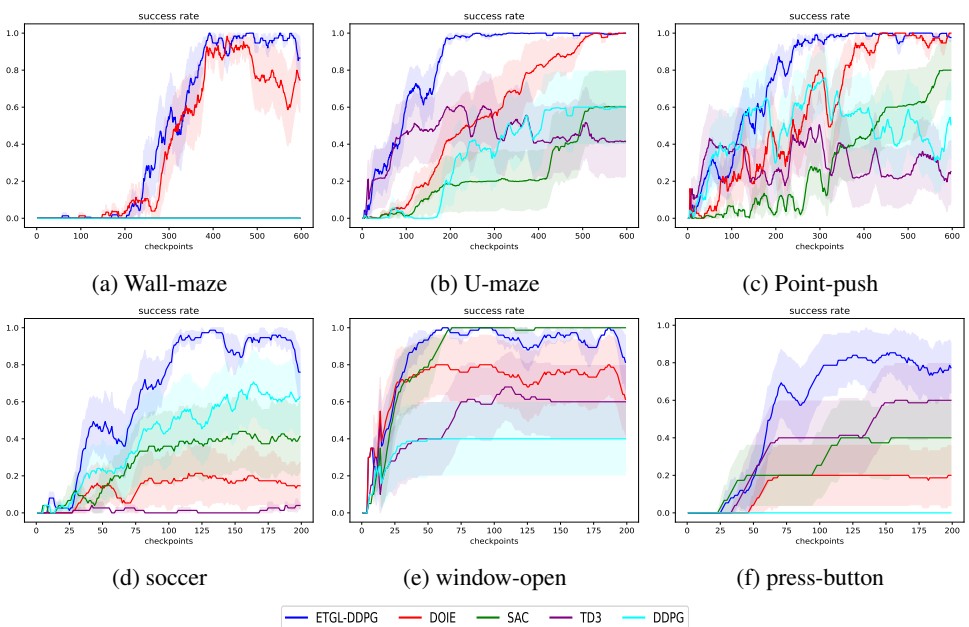

Figure 3: The success rates across all environments, averaged over 5 runs with different random seeds. Shaded areas represent one standard deviation. We trained all methods for 6 million frames in the navigation environments and 2 million frames in the manipulation environments, with success rates reported at every $10^5$-step checkpoint. A moving average with a window size of 10 is applied to all methods for better readability.

ronments are *U-maze* (Figure 2b) and *Point-push* (Figure 2c) (Kanagawa, 2021), designed using the MuJoCo physics engine (Todorov et al., 2012). In both environments, a robot (orange ball) seeks to reach the goal (red region). In Point-push, the robot must additionally push aside the two movable red blocks that obstruct the path to the goal. A small negative reward of -0.001 is given at each step unless the goal is reached, where the reward is 1. In each episode, the robot starts near the same position with slight random variations, but the goal region remains fixed.

We also employ three manipulation tasks: *window-open*, *soccer*, and *button-press* (Figures 2d, e, and f) (Yu et al., 2020). In window-open, the goal is to push the window open; in soccer, the goal is to kick the ball into the goal; and in button-press, the aim is to press the top-down button. Each episode begins with the robot's gripper in a randomized starting position, while the positions of other objects remain constant. The original versions of these tasks employ a uniquely shaped reward function for each task. However, these versions offer limited challenges for exploration, as standard baselines, such as SAC, demonstrate strong performance (Yu et al., 2020). We modified the original reward function to be sparse, transforming these tasks into challenging exploration problems.

The maximum number of steps per episode is set to 100 for Wall-maze and 500 for all other environments. Across all methods, the neural network architecture consists of 3 hidden layers with 128 units each, using ReLU activation functions. For standard baselines, we utilize the implementations from OpenAI Gym (Dhariwal et al., 2017), and for other baselines, we rely on their publicly available

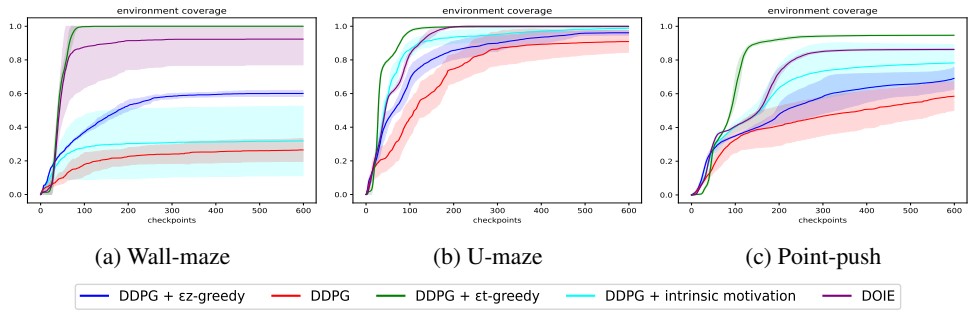

Figure 4: The environment coverage for exploration strategies in navigation environments. On the graph, the y-axis indicates the portion of the environment that has been covered, and the checkpoints occur every $10^4$ steps shown on the x-axis. The results are given for the average of 10 runs with random seeds. The shaded region represents one standard deviation.

implementations. After testing various configurations, we found that $\epsilon t$-greedy and $\epsilon z$-greedy perform best with budgets of $N = 40$ and $N = 15$, respectively, across these environments. Additional details about the environments and experimental setup are provided in Appendix A.3.

## 4.1 OVERALL PERFORMANCE OF ETGL-DDPG

We evaluate the performance of ETGL-DDPG compared to state-of-the-art methods. We compare with SAC (Haarnoja et al., 2018), TD3 (Fujimoto et al., 2018), DDPG, and DOIE (Lobel et al., 2022). DOIE demonstrates state-of-the-art performance in challenging sparse-reward continuous control problems by drastically improving the exploration. While both DOIE and $\epsilon t$-greedy use a similarity measure between new and observed states, DOIE applies this to compute an optimistic value function rather than solely guiding the agent to unexplored areas. The results are shown in Figure 3. In the navigation environments, ETGL-DDPG and DOIE demonstrate superior performance compared to other methods, with ETGL-DDPG achieving a success rate of 1 faster than DOIE. Notably, Wall-maze presents a more challenging task among navigation environments, where only ETGL-DDPG and DOIE are able to achieve a success rate above zero. In manipulation tasks, the press-button poses the hardest challenge as none of the methods achieve a success rate of 1. ETGL-DDPG still outperforms all other approaches, while DOIE underperforms compared to SAC, indicating its limitations in adapting to high dimensional environments.

## 4.2 ENVIRONMENT COVERAGE THROUGH EXPLORATION

We now examine how effective $\epsilon t$-greedy is in covering the environment. To do so, we discretize the navigation environments into small cells. A cell is considered visited if the agent encounters a sufficient number of distinct states within it, and the overall environment coverage is quantified as the fraction of visited cells. Figure 4 presents a comparison of environment coverage across different exploration strategies. All strategies except DOIE, which uses Radial Basis Function Deep Q-Network (RBFDQN) (Asadi et al., 2021), use DDPG as their underlying algorithm. RBFDQN is an enhanced DQN variant that incorporates Radial Basis Functions (RBF) to achieve more accurate Q-value approximations in continuous environments. In Wall-maze, $\epsilon t$-greedy is the only method capable of fully covering the environment, while DOIE achieves 90% coverage. $\epsilon z$-greedy covers approximately half of the environment, whereas the remaining methods manage to explore only around 30%. In U-maze, all strategies are successful, covering 80% or more of the environment. Even so, both $\epsilon t$-greedy and DOIE reach full coverage faster than other methods. In Point-push, none of the methods can fully cover the environment. However, $\epsilon t$-greedy still outperforms all baselines, and among the baselines, DOIE explores more than the others. We also investigate the distribution of final states reached in the episodes to determine the order in which the agent visits different regions of the environment (see Appendix A.5).

The tree budget $N$ upper bounds the option length of $\epsilon t$-greedy due to the fact that the longest path between nodes in the tree is shorter or equal to the number of nodes in the tree. This is analogous to

Table 1: Analysis of the impact of budget N on the environment coverage.

| budget $N$ | $\epsilon z$-greedy | | | $\epsilon t$-greedy | | |
|---|---|---|---|---|---|---|
| | Wall-maze | U-maze | Point-push | Wall-maze | U-maze | Point-push |
| 5 | 0.36 | 0.55 | 0.36 | 0.76 | 0.94 | 0.40 |
| 10 | **0.38** | 0.91 | 0.38 | 0.97 | 0.91 | 0.41 |
| 15 | 0.34 | 0.85 | 0.39 | 0.65 | 0.94 | 0.42 |
| 20 | 0.30 | 0.84 | 0.40 | 0.83 | 0.94 | 0.48 |
| 25 | 0.28 | **0.86** | 0.40 | 1 | 0.95 | 0.47 |
| 30 | 0.27 | 0.83 | 0.39 | 1 | **0.97** | 0.51 |
| 35 | 0.25 | 0.82 | 0.40 | 1 | 0.95 | 0.53 |
| 40 | 0.24 | 0.82 | 0.40 | 1 | 0.97 | 0.55 |
| 45 | 0.22 | 0.85 | **0.41** | 1 | 0.96 | 0.64 |
| 50 | 0.22 | 0.79 | 0.40 | 1 | 0.97 | **0.73** |

the role of $N$ in $\epsilon z$-greedy, where a uniform distribution $z(n) = \mathbb{1}_{n \leq N}/N$ is used. To evaluate both methods, we assess environment coverage under varying budget sizes, calculating the coverage after 1 million training frames. Table 1 shows the results: $\epsilon t$-greedy consistently achieves greater coverage than $\epsilon z$-greedy across all environments and budget sizes. Additionally, $\epsilon t$-greedy demonstrates improved the coverage as the budget increases. In contrast, increasing the budget for $\epsilon z$-greedy does not consistently improve coverage and can even decrease it in some cases. This highlights the advantages of directed exploration over undirected methods, particularly in complex environments with numerous obstacles, such as Wall-maze.

### 4.3 EFFECTIVENESS OF EACH NEW COMPONENT IN ETGL-DDPG

We evaluated the performance of ETGL-DDPG, and now we assess the impact of each component on DDPG separately. Figure 5 presents the results for all environments. $\epsilon t$-greedy demonstrates the most improvement across all environments and is the only method that enhances the performance of DDPG in the Wall-maze, highlighting the critical role of our exploration strategy. GDRB shows a positive impact on DDPG performance in all environments, except for soccer, where DDPG alone outperforms all baselines. Additionally, we replaced reservoir sampling with FIFO as the retention policy in GDRB and observed similar results. The longest n-step return has a positive effect only in U-maze and press-button tasks, while it negatively impacts performance in soccer and Point-push. We attribute this to the inherently high variance of multi-step updates. A comparison of Figures 3 and 5 across all environments shows that ETGL-DDPG consistently outperforms the use of each component individually, supporting the effectiveness of their combination.

### 5 RELATED WORK

**Exploration.** Intrinsic motivation methods (Burda et al., 2018; Pathak et al., 2017; Ostrovski et al., 2017; Tang et al., 2017) provide a reward bonus for unexplored areas of the state space. These methods make the reward function non-stationary, which breaks the Markov assumption of MDP. Decoupled RL algorithms (Schäfer et al., 2021; Badia et al., 2019) resolve the non-stationarity of the reward function by training two separate policies for exploration and exploitation. However, such methods require double the computation cost. Colas et al. (2018) use a policy search process to generate diverse data for training of DDPG. Liu et al. (2018) introduce a competition-based exploration method where two agents (A and B) compete with each other. Agent A is penalized for visiting states visited by B, while B is rewarded for visiting states discovered by A. Plappert et al. (2018) directly inject noise into the policy's parameter space instead of the action space. Eysenbach et al. (2019) build a graph using states in the replay buffer, allowing the agent to navigate distant regions of the environment by applying Dijkstra's algorithm. Lobel et al. (2022) present Deep Optimistic Initialization for Exploration (DOIE), which improves exploration in continuous control tasks by maintaining optimism in state-action value estimates. Lobel et al. (2023) demonstrate that DOIE can estimate visit counts by averaging samples from the Rademacher distribution instead of using density models. Dey et al. (2024) present COIN, a continual optimistic initialization strategy that extends DOIE to stochastic and non-stationary environments.

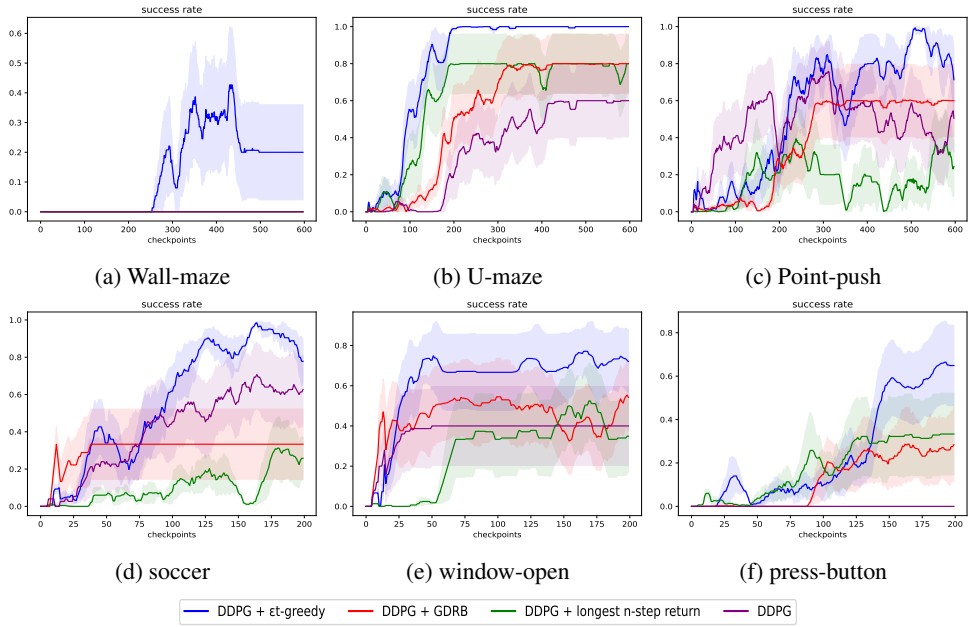

(a) Wall-maze      (b) U-maze      (c) Point-push

(d) soccer      (e) window-open      (f) press-button

Figure 5: Analyzing the individual impact of three components on DDPG: $\epsilon t$-greedy, GDRB, and longest n-step return.

**Experience Replay Buffer and Reward Propagation.** Rather than uniformly sampling from the buffer, Prioritized Experience Replay (PER) (Schaul et al., 2015) prioritizes transitions in the buffer based on reward, recency, or TD error at the expense of $O(\log N)$ per sample, where $N$ is the buffer size. CER (Zhang & Sutton, 2017) includes the last transition from the buffer to each sampled batch with $O(1)$ complexity. Zhang et al. (2022) learn a conservative value regularizer only from the observed transitions in the replay buffer to improve the sample efficiency of DQN. Pan et al. (2022) theoretically show why PER has a better convergence rate than uniform sampling policy when minimizing mean squared error. Furthermore, Pan et al. (2022) identify two limitations of PER: outdated priorities and insufficient coverage of the state space. Reward shaping (Laud, 2004; Hu et al., 2020) creates artificial intermediate rewards to facilitate reward propagation. However, designing appropriate intermediate rewards is hard and often problem-specific. Trott et al. (2019) address this issue by introducing *self-balancing reward shaping* in the context of on-policy learning. To extract more information from an unsuccessful episode, Andrychowicz et al. (2017) introduce *imaginary goals*. An imaginary goal for state $s$ is a state that is encountered later in the episode. Devidze et al. (2024) introduce a novel reward informativeness criterion that adaptively designs interpretable reward functions based on an agent's current policy in sparse-reward tasks.

## 6 CONCLUSIONS AND FUTURE WORK

We have introduced the ETGL-DDPG algorithm with three components that improve the performance of DDPG for sparse-reward goal-conditioned environments. $\epsilon t$-greedy is a temporally-extended version of $\epsilon$-greedy using options generated by search. We prove that $\epsilon t$-greedy achieves a polynomial sample complexity under specific MDP structural assumptions. GDRB employs an extra buffer to separate successful episodes. The longest $n$-step return bootstraps from the Q-value of the final state in unsuccessful episodes and becomes a Monte Carlo update in successful episodes. ETGL-DDPG uses these components with DDPG and outperforms state-of-the-art methods, at the expense of about 1.5x wall-clock time w.r.t DDPG. The current limitation of our work is that we approximate visit counts through static hashing. For image-based problems such as real-world navigation, the future direction is to leverage dynamic hashing techniques such as *normalizing flows* (Papamakarios et al., 2021) as these tasks demand more intricate representation learning.

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

# A  APPENDIX

## A.1  $\epsilon t$-GREEDY SAMPLE EFFICIENCY : PROOFS

In this section, we first provide an overview of the proof, presenting the key ideas at a high level. Then, we present the detailed formal proof of Theorem 1 and Theorem 2.

**Proof Overview.**  We aim to show that the $\epsilon t$-greedy algorithm falls into the PAC-MDP category. According to Liu & Brunskill (2018), an algorithm $\mathcal{A}$ is PAC-MDP if the covering time induced by $\mathcal{A}$ is polynomially bounded. In Liu & Brunskill (2018), the authors further demonstrate that bounding the covering time requires bounding both the Laplacian eigenvalues and the stationary distribution over the states induced by the random walk policy. This is presented as Proposition A.1. According to Theorem 2, two conditions are satisfied: $N \leq \Theta(|\mathcal{S}||\mathcal{A}|)$ and a lower bound on the probability of the sampled option, $\mathcal{P}_{\mathcal{X}} \geq \frac{1}{\Theta(|\mathcal{S}||\mathcal{A}|)}$. These two conditions are necessary and are met by our problem setting and the exploration algorithm (Algorithm 1). To prove that $\mathcal{P}_{\mathcal{X}} \geq \frac{1}{\Theta(|\mathcal{S}||\mathcal{A}|)}$, we construct a worst-case tree structure $\mathcal{X}$, where we aim to identify the option induced by the tree $\mathcal{X}$ with the lowest probability, referred to informally as the "hardest option". We then show that this lower bound satisfies the condition specified in Theorem 1.

We now proceed with the proof of Theorem 1, as demonstrated below.

**Theorem 1** (**Worst-Case Sampling**).  *Given a tree $\mathcal{X}$ with $N$ nodes ($s_1$ to $s_N$), for any $\omega \in \Omega_{\mathcal{X}}$, the sampling probability satisfies:*

$$\mathcal{P}_{\mathcal{X}}[\omega] \geq \frac{1}{N!(\max_{i \in [N]} |\phi(s_i)|)^{N-1}} \geq \frac{1}{\Theta(|\mathcal{S}||\mathcal{A}|)} \tag{7}$$

*where $N \leq \frac{\log(|\mathcal{S}||\mathcal{A}|)}{\log\log(|\mathcal{S}||\mathcal{A}|)}$ Here, $\mathcal{S}$ and $\mathcal{A}$ represent the state space and action space, respectively.*

*Proof.*  As outlined in the proof overview, we need to construct an option with the lowest sampling probability. Given a tree $\mathcal{X}$, we define $\mathcal{X}_i$ (for $1 \leq i \leq N$) as the tree constructed up to the $i$-th time step. At each step $\mathcal{X}_i$, we track the tuple of added states, denoted by $\mathcal{S}_i^{\mathcal{X}}$, the uniformly sampled state $s_x$ from $\mathcal{S}_i^{\mathcal{X}}$, and the state with the fewest visits, $s_{min}$. The notation $s_x$ and $s_{min}$ follows Algorithm 1. Without loss of generality, we assume that each next state $s_{x'}$ in line 9 of Algorithm 1 satisfies $n(\phi(s_{x'})) \neq 0$. Specifically, we consider a worst-case tree $\mathcal{X}$ fully populated with states from $s_1$ to $s_N$. Therefore, at time step $N$, $\mathcal{S}_N^{\mathcal{X}} = (s_1, s_2, \ldots, s_N)$, and we have the following relation:

$$n(\phi(s_1)) \geq n(\phi(s_2)) \geq n(\phi(s_3)) \cdots \geq n(\phi(s_N)). \tag{8}$$

Equation 8 provides a decreasing sequence of visitations for newly added nodes in tree $\mathcal{X}$, emphasizing line 15 of Algorithm 1, which causes the state $s_{min}$ to change over $N$ iterations. We assume a specific structure for each $\phi(s_i)$, where for all $i \in [N]$, at each bucket $\phi(s_i)$, there exists only one state denoted by $s_{i+1}$, such that $n(\phi(s_{i+1})) \leq n(\phi(s_i))$. Additionally, we assume that at each time step in $\mathcal{X}_t$, the newly added node connects only to the most recently added node in the tree. The two key stochastic events are summarized as follows:

- $\mathcal{E}_1$: The event in which nodes are sampled in Line 24 from buckets satisfying the increasing sequence above.

- $\mathcal{E}_2$: The event in which nodes are selected in Line 8.

We now define the probability of interest, which we aim to bound:

$$\mathcal{P}[\text{option returned from } s_{\text{root}} \text{ to } s_N | \mathcal{E}_1 \text{ and } \mathcal{E}_2]. \tag{9}$$

We expand this probability as follows:

$$\mathcal{P}[\text{option returned from } s_{\text{root}} \text{ to } s_N \mid \mathcal{E}_1 \text{ and } \mathcal{E}_2] = \prod_{i=2}^{N} \mathcal{P}[(\text{State } s_i \text{ added to tree } \mathcal{X}) \wedge (s_i = s_{min}) \wedge (s_x = s_{i-1} \text{ in Line 8})]$$

$$= \prod_{i=2}^{N} \frac{1}{(i-1)|\phi(s_{i-1})|}$$

$$= \frac{1}{(N-1)!} \times \frac{1}{|\phi(s_1)||\phi(s_2)| \ldots |\phi(s_N)|}$$

$$> \frac{1}{N!} \times \frac{1}{(\max_{i \in [N]} |\phi(s_i)|)^{N-1}}$$

$$> \frac{1}{|\mathcal{S}||\mathcal{A}|}.$$

To prove the final inequality, note that $N \leq \frac{\log(|\mathcal{S}||\mathcal{A}|)}{\log\log(|\mathcal{S}||\mathcal{A}|)}$. Since the size of the sets $\mathcal{S}$ and $\mathcal{A}$ is large and $N$ is sub-logarithmic in $|\mathcal{S}||\mathcal{A}|$, i.e., $N \ll \log(|\mathcal{S}||\mathcal{A}|)$, we can say $N \leq \frac{\log(|\mathcal{S}||\mathcal{A}|)}{\log(N)}$. Let us denote $\log(\max_{i \in [N]} |\phi(s_i)|)$ as a constant $c_0$.

Now by the series of following inequalities we prove that $\frac{1}{N!} \times \frac{1}{(\max_{i \in [N]} |\phi(s_i)|)^{N-1}} > \frac{1}{|\mathcal{S}||\mathcal{A}|}$.

$$N \leq \frac{\log(|\mathcal{S}||\mathcal{A}|)}{\log(N)} \Rightarrow N \log(N) \leq \log(|\mathcal{S}||\mathcal{A}|) \tag{10}$$

$$\Rightarrow N \log(N) + (N-1)c_0 - N \leq \log(|\mathcal{S}||\mathcal{A}|) \qquad (\text{since } |\mathcal{S}||\mathcal{A}| \gg N, c_0) \tag{11}$$

$$\Rightarrow \log(N!) + (N-1)c_0 \leq \log(|\mathcal{S}||\mathcal{A}|) \qquad (\text{Based on the Moivre–Stirling approximation}) \tag{12}$$

$$\Rightarrow \log(N!) + (N-1)c_0 \leq \log(|\mathcal{S}||\mathcal{A}|) \tag{13}$$

$$\Rightarrow \log(N!) + \log\left(\left(\max_{i \in [N]} |\phi(s_i)|\right)^{N-1}\right) \leq \log(|\mathcal{S}||\mathcal{A}|) \tag{14}$$

$$\Rightarrow \log\left(N! \cdot \left(\max_{i \in [N]} |\phi(s_i)|\right)^{N-1}\right) \leq \log(|\mathcal{S}||\mathcal{A}|) \tag{15}$$

$$\Rightarrow \frac{1}{N! \cdot (\max_{i \in [N]} |\phi(s_i)|)^{N-1}} \geq \frac{1}{|\mathcal{S}||\mathcal{A}|} \tag{16}$$

$$\square$$

Now we provide the main proof which demonstrates polynomial sample complexity under certain criteria.

**Theorem 2** ($\epsilon t$-greedy Sample Efficiency). *Given a state space $\mathcal{S}$, action space $\mathcal{A}$, and a set of options $\Omega_{\mathcal{X}}$ generated by $\epsilon t$-greedy for each tree $\mathcal{X}$, if $\mathcal{P}_{\mathcal{X}}[\omega] \geq \frac{1}{\Theta(|\mathcal{S}||\mathcal{A}|)}$, $\epsilon t$-greedy achieves polynomial sample complexity or i.e. is PAC-MDP.*

*Proof.* First note that if $\mathcal{P}_{\mathcal{X}}[\omega] \geq \frac{1}{\Theta(|\mathcal{S}||\mathcal{A}|)}$ then based on Theorem 1 we need to have $N \leq \frac{\log(|\mathcal{S}||\mathcal{A}|)}{\log\log(|\mathcal{S}||\mathcal{A}|)}$, and this implies that $N \leq \Theta(|\mathcal{S}||\mathcal{A}|)$. Based on the paper by (Liu & Brunskill, 2018), and the analysis of the covering length when following a random policy, we have the following preposition:

**Preposition A.1** (**Liu & Brunskill (2018)**). *: For any irreducable MDP $M$, we define $P_{\pi_{RW}}$ as a transition matrix induced by random walk policy $\pi_{RW}$ over $M$ and $L(P_{\pi_{RW}})$ is denoted as the Laplacian of this transition matrix. Suppose $\lambda$ is the smallest non-zero eigenvalue of $L$ and $\Psi(s)$ is the stationary distribution over states which is induced by random walk policy, then Q-learning with random walk exploration is a PAC RL algorithm if: $\frac{1}{\lambda}, \frac{1}{\min_s \Psi(s)}$ are Poly($|\mathcal{S}||\mathcal{A}|$).*

Note that Preposition A.1 is not limited to an MDP with primitive actions. Therefore, we can broaden its scope by incorporating options into this proposition and demonstrate that both $\frac{1}{\lambda}$ and $\frac{1}{\min_s \Psi(s)}$ can be polynomially bounded in terms of MDP parameters—in this case, states and actions in our approach.

Let's begin by examining the upper-bound for $\frac{1}{\min_s \Psi(s)}$. Suppose we are at exploration tree $\mathcal{X}$. Without a loss of generality, we consider that capacity of tree $\mathcal{X}$ is full, and we have $N$ states. In this tree, let's designate $s_{root}$ as the state assigned as the root of the tree during the exploration phase. Now, consider another random state (excluding $s_{root}$) within this tree structure, denoted as $s_{rand}$. We acknowledge that, when considering the entire state space, there can be multiple options constructed from $s_{root}$ to $s_{rand}$. Each tree $\mathcal{X}$ provides one of these options. $\Psi(s)$ is defined over all states, and $\omega$ is the option with a limited size because of the constrained tree budget.

we can calculate the upper-bound for $\frac{1}{\min_s \Psi(s)}$ as follows:

$$\Psi(s_{rand}) = \sum_{\omega \in \Omega_{\mathcal{X}}} \mathcal{P}_{\mathcal{X}}[\omega]\Psi(s_{root}) \Rightarrow \Psi(s_{rand}) \geq \mathcal{P}[\omega]\Psi(s_{root}),$$

$$\frac{1}{\Psi(s_{rand})} \leq \frac{1}{\mathcal{P}[\omega]}\frac{1}{\Psi(s_{root})} \Rightarrow \frac{1}{\Psi(s_{rand})} \leq \frac{\Theta(|\mathcal{S}||\mathcal{A}|)}{\Psi(s_{root})} \tag{17}$$

Since $s_{rand}$ can represent any of the states encountered in the tree, we can regard it as the state assigned the least probability in the stationary distribution. Therefore, we have:

$$\frac{1}{\Psi(s_{rand})} \leq \frac{\Theta(|\mathcal{S}||\mathcal{A}|)}{\Psi(s_{root})} \Rightarrow \frac{1}{\min_s \Psi(s)} \leq \frac{\Theta(|\mathcal{S}||\mathcal{A}|)}{\Psi(s_{root})} \tag{18}$$

So, $\frac{1}{\min_s \Psi(s)}$ is polynomially bounded. Now, we need to demonstrate that $\frac{1}{\lambda}$ is also polynomially bounded. To bound $\lambda$, we first need to recall the definition of the Cheeger constant, $h$. Drawing from graph theory, if we denote $V(G)$ and $E(G)$ as the set of vertices and edges of an undirected graph $G$, respectively, and considering the subset of vertices denoted by $V_s$, we can define $\sigma V_s$ as follows:

$$\sigma V_s := \{(n_1, n_2) \in E(G) : n_1 \in V_s, n_2 \in V(G) \setminus V_s\} \tag{19}$$

So, $\sigma V_s$ can be regarded as a collection of all edges going from $V_s$ to the vertex set outside of $V_s$. In the above definition, $(n_1, n_2)$ is considered as a graph edge. Now, we can define a Cheeger constant:

$$h(G) := \min\{\frac{|\sigma V_s|}{|V_s|} : V_s \subseteq V(G), 0 < V_s \leq \frac{1}{2}|V(G)|\} \tag{20}$$

We are aware that $h \geq \lambda \geq \frac{h^2}{2}$, and by polynomially bounding $h$, we can ensure that $\lambda$ is also bounded. In a related work (Liu & Brunskill, 2018), an alternative variation of the Cheeger constant is utilized, which is based on the flow $F$ induced by the stationary distribution $\Psi$ of a random walk on the graph. Suppose for nodes $n_1, n_2$ and subset of nodes $N_1$ in the graph, we have:

$$F(n_1, n_2) = \Psi(n_1)P(n_1, n_2), \tag{21}$$

$$F(\sigma N_1) = \sum_{n_1 \in N_1, n_2 \notin N_1} F(n_1, n_2), \tag{22}$$

$$F(N_1) = \sum_{n_1 \in N_1} \Psi(n_1) \tag{23}$$

Building upon the aforementioned definition, the Cheeger constant is defined as:

$$h := \inf_{N_1} \frac{F(\sigma N_1)}{\min\{F(N_1), F(\bar{N}_1)\}} \tag{24}$$

Suppose $N_{rand} = \{s_{root}\}$; we will now demonstrate that $\frac{1}{h}$ can be polynomially bounded :

$$h = \inf_{N_1} \frac{F(\sigma N_1)}{\min\{F(N_1), F(\bar{N}_1)\}} \geq \frac{F(\sigma N_{rand})}{\min\{F(N_{rand}), F(\overline{N_{rand}})\}} \geq \frac{\sum_{s \neq s_{root}} \Psi(s_{root}) P_{\pi_{RW}}(s_{root}, s)}{\Psi(s_{root})},$$

$$= \sum_{s \neq S_{root}} P_{\pi_{RW}}(s_{root}, s) \geq \mathcal{P}_{\mathcal{X}}[\omega] \Rightarrow \frac{1}{h} \leq \Theta(|\mathcal{S}||\mathcal{A}|)$$

We demonstrate that both terms stated in Preposition A.1 are polynomially bounded, and thus, the proof is complete. $\qquad\square$

---

**Algorithm 2** Generating exploratory option with tree search using a perfect model

---

1: **function generate_option**(state s, hash function $\phi$, budget N)
2:     frontier_nodes ← {}
3:     Initialize root using $s$: $root \leftarrow TreeNode(s)$
4:     frontier_nodes ← frontier_nodes ∪ {root};
5:     $s_{\min} \leftarrow$ root
6:     $i \leftarrow 0$
7:     **while** $i < N$ **do**
8:         $s_x \sim UniformRandom$(frontier_nodes)
9:         $s_{x'}$ = **next_state_from_env**($s_x$)
10:        **if** $\phi(n(s_{x'}))$=0 **then**
11:           Extract option $o$ by actions $root$ to $s_{x'}$
12:           **return** $o$
13:        **end if**
14:        **if** $n(\phi(s_{x'})) < n(\phi(s_{\min}))$ **then**
15:           $s_{\min}$=$s_{x'}$
16:        **end if**
17:        $i \leftarrow i + 1$
18:     **end while**
19:     Extract option $o$ by actions $root$ to $s_{\min}$
20:     **return** $o$
21: **end function**
22:
23: **function next_state_from_env**($s_x$, frontier_nodes)
24:     a $\sim UniformRandom(\mathcal{A}(s_x))$
25:     $s_{x'} \leftarrow \mathcal{T}(s_x, a)$
26:     $s_x$.add_child ($s_{x'}$)
27:     frontier_nodes ← frontier_nodes ∪ $\{s_{x'}\}$
28:     **return** $s_{x'}$
29: **end function**

---

### A.2 EXPLORATION WITH A PERFECT MODEL

Since the DDPG algorithm is model-free, we utilize the replay buffer to construct the tree for $\epsilon t$-greedy. However, $\epsilon t$-greedy can also take advantage of a perfect model when available. The pseudocode for option generation using a perfect model is provided in Algorithm 2. The key difference from Algorithm 1 is the use of the `next_state_from_env` function instead of `next_state_from_replay_buffer` to generate child nodes. In this case, an action is uniformly sampled from the action space, and the environment's transition function $\mathcal{T}$ is directly used to determine the next state (line 25). Figure 6 compares the performance of ETGL-DDPG in navigation environments using a perfect model versus a replay buffer. The results show a clear advantage when using a perfect model, as the agent reaches a success rate of 1 more quickly and with less deviation.

### A.3 IMPLEMENTATION DETAILS AND EXPERIMENTAL HYPERPARAMETERS

Here, we describe the implementation details and hyperparameters for all methods used in this paper. All experiments were run on a system with 5 vCPU on a cluster of Intel Xeon E5-2650 v4

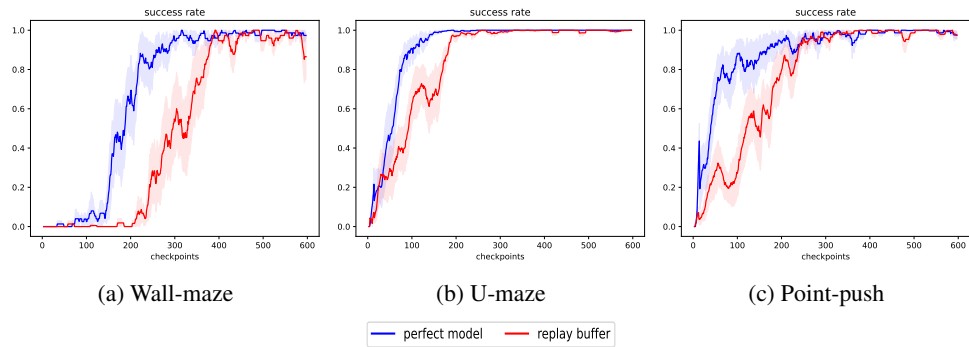

Figure 6: Comparison of ETGL-DDPG performance in navigation environments using a perfect model vs. replay buffer.

2.2GHz CPUs and one 2080Ti GPU. Table 3 displays the details for environments. Tables 2, 4, and 5 showcase the hyperparameters utilized in ETGL-DDPG and the baselines.

Table 2: Implementation details for ETGL-DDPG.

| Hyperparameter | wall-maze | U-maze | Point-push | window-open | soccer | button-press |
|---|---|---|---|---|---|---|
| batch size | | 128 | | | 512 | |
| number of updates per episode | | 20 | | | 200 | |
| epsilon decay rate | | 0.9999988 | | | 0.9999992 | |
| exploration budget $N$ | 20 | | 40 | | 60 | |
| SimHash dimension | | $k = 9$ | | | $k = 16$ | |
| soft target updates $\tau$ | | | $10^{-2}$ | | | |
| discount factor $\gamma$ | | | 0.99 | | | |
| warmup period | | | $2 * 10^5$ steps | | | |
| exploration buffer size | | | $10^6$ | | | |
| exploitation buffer size | | | $5 * 10^4$ | | | |
| actor learning rate | | | $10^{-4}$ | | | |
| critic learning rate | | | $10^{-3}$ | | | |

Table 3: Environment details.

| environment | $S \in$ | $G \in$ | $A \in$ | Max steps per episode |
|---|---|---|---|---|
| Wall-maze | $\mathbb{R}^2$ | $\mathbb{R}^2$ | $[-0.95, 0.95]^2$ | 100 |
| U-maze | $\mathbb{R}^6$ | $\mathbb{R}^2$ | $[-1, 1] * [-0.25, 0.25]$ | 500 |
| Point-push | $\mathbb{R}^{11}$ | $\mathbb{R}^2$ | $[-1, 1] * [-0.25, 0.25]$ | 500 |
| window-open | $\mathbb{R}^{39}$ | $\mathbb{R}^3$ | $[-1, 1]^4$ | 500 |
| soccer | $\mathbb{R}^{39}$ | $\mathbb{R}^3$ | $[-1, 1]^4$ | 500 |
| button-press | $\mathbb{R}^{39}$ | $\mathbb{R}^3$ | $[-1, 1]^4$ | 500 |

Table 4: Implementation details for SAC, TD3, and DDPG.

| Hyperparameter | wall-maze | U-maze | Point-push | window-open | soccer | button-press |
|---|---|---|---|---|---|---|
| batch size | 128 | | | 512 | | |
| update frequency per step | 12 | | | 2 | | |
| action noise | $\sim N(0, 0.2)$ | $\sim N(0, (0.3, 0.05))$ | | $\sim N(0, (0.15))$ | | |
| warmup period | $2 * 10^5$ steps | | | | | |
| replay buffer size | $10^6$ | | | | | |
| learning rate | $3 * 10^{-4}$ | | | | | |
| soft target updates $\tau$ | $5 * 10^{-3}$ | | | | | |
| discount factor $\gamma$ | 0.99 | | | | | |

Table 5: Implementation details for DOIE.

| Hyperparameter | wall-maze | U-maze | Point-push | window-open | soccer | button-press |
|---|---|---|---|---|---|---|
| batch size | 256 | | | | | |
| number of updates per episode | 100 | | | | | |
| replay buffer size | $5 * 10^5$ | | | | | |
| actor learning rate | $10^{-4}$ | | | | | |
| critic learning rate | $5 * 10^{-3}$ | | | | | |
| discount factor $\gamma$ | 0.99 | | | | | |
| action scaling | 0.01 | | | | | |
| environment scaling | 0.1 for each dimension | | | | | |
| knownness mapping type | polynomial | | | | | |

## A.4 ETGL-DDPG ALGORITHM

In this section, we introduce ETGL-DDPG, as detailed in Algorithm 3, which is organized into three primary functions: `train`, `run_episode`, and `update`. The `train` function is called once at the start of the training process. For each training episode, the `run_episode` function is invoked to perform a training episode within the environment, followed by the `update` function to adjust the networks based on the experience gained from the episode.

## A.5 TERMINAL STATES DISTRIBUTION

We analyze the order in which the agent visits different parts of the environment by examining the distribution of the last states in the episodes. To make it more visually appealing and easy to interpret, we only sample some of the episodes. The results for Wall-maze, U-maze, and Point-push can be found in Figures 7, 8, and 9, respectively. In Wall-maze, only $\epsilon t$-greedy and DOIE can effectively navigate to different regions of the environment and ultimately reach the goal area. Other methods often get trapped in one of the local optima and are unable to reach the goal. The reason some methods, such as TD3, have fewer points is that the agent spends a lot of time revisiting congested areas instead of exploring new ones. In U-maze, most methods can explore the majority of the environment. However, during the final stages of training, methods such as DDPG, SAC, and DDPG + intrinsic motivation have lower success rates and may end up in locations other than the goal areas. In Point-push, $\epsilon t$-greedy, $\epsilon z$-greedy, and DOIE first visit the lower section of the environment in the early stages. After that, they push aside the movable box and proceed to the upper section to visit the goal area. For the other methods, the pattern is almost the same, with occasional visits to the lower section.

---

**Algorithm 3** ETGL-DDPG

---

Randomly initialize critic network $Q(s, a, g|\theta^Q)$ and actor $\mu(s, g|\theta^\mu)$ with weights $\theta^Q$ and $\theta^\mu$

Initialize target networks $Q'$ and $\mu'$ with weights $\theta^{Q'} \leftarrow \theta^Q$, $\theta^{\mu'} \leftarrow \theta^\mu$

Initialize replay buffers $D_\beta$, $D_e$, hash function $\phi$, exploration budget $N$

**function train**($Q, \mu, \phi$)
    **for** episodes=1,M **do**
        Receive initial observation state $s_1$ and goal $g$
        **run_episode**($s_1, g$)
        **update**($success$)
    **end for**
**end function**

**function run_episode**($s, g$)
    $success \leftarrow false, l \leftarrow 0$
    **while** t $\leq$ T **and not**($success$) **do**
        **if** $l$==0 **then**
            **if** random()$< \epsilon$ **then**
                Exploratory option $w \leftarrow$ **generate_option**($s, \phi, N$)
                Assign action : $a_t \leftarrow w$
                $l \leftarrow$ length($w$)
            **else**
                Greedy action : $a_t \leftarrow \mu(s_t, g|\theta^\mu)$
            **end if**
        **else**
            Assign action : $a_t \leftarrow w$
            $l \leftarrow l - 1$
        **end if**
        Execute action $a_t$ and observe reward $r_t$ and next state $s_{t+1}$
        **if** is_goal($s_{t+1}$) **then**
            $success \leftarrow true$
        **end if**
    **end while**
**end function**

**function update**($success$)

$$R = \begin{cases} r_t & success \\ 0 & o.w \end{cases} \quad bootstrap = \begin{cases} 0 & success \\ 1 & o.w \end{cases}$$

    **for** $i \in \{t-1, ..., t_{start}\}$ **do**
        $R \leftarrow r_i + \gamma R$
        **if** $success$ **then**
            store transition $(s_i, g, a_i, R, s_t, bootstrap)$ in $D_\beta, D_e$
        **else**
            store transition $(s_i, g, a_i, R, s_t, bootstrap)$ in $D_\beta$
        **end if**
    **end for**
    Sample C random mini-batches of k transitions $(s_j, g_j, a_j, r_j, s_{j+1}, bootstrap_j)$ by $\tau_\beta$ and $\tau_e$ ratios from $D_\beta$ and $D_e$
    set $y_j = r_j + bootstrap_j * \gamma Q'(s_{j+1}, g_j, \mu'(s_{j+1}, g_j|\theta^{\mu'})|\theta^{Q'})$
    update critic by minimizing the loss: $L = \frac{1}{k}\sum_j(y_j - Q(s_j, g_j, a_j|\theta^Q))$
    update the actor: $\nabla_{\theta^\mu} J \approx \frac{1}{k}\sum_j \nabla_a Q(s, g, a|\theta^Q)|_{s=s_j, g=g_j, a=\mu(s_j, g_j)} \nabla_{\theta^\mu}\mu(s, g|\theta^\mu)|_{s_j}$
    update the target networks: $\theta^{Q'} \leftarrow \tau\theta^Q + (1-\tau)\theta^{Q'}$, $\theta^{\mu'} \leftarrow \tau\theta^\mu + (1-\tau)\theta^{\mu'}$
**end function**

---

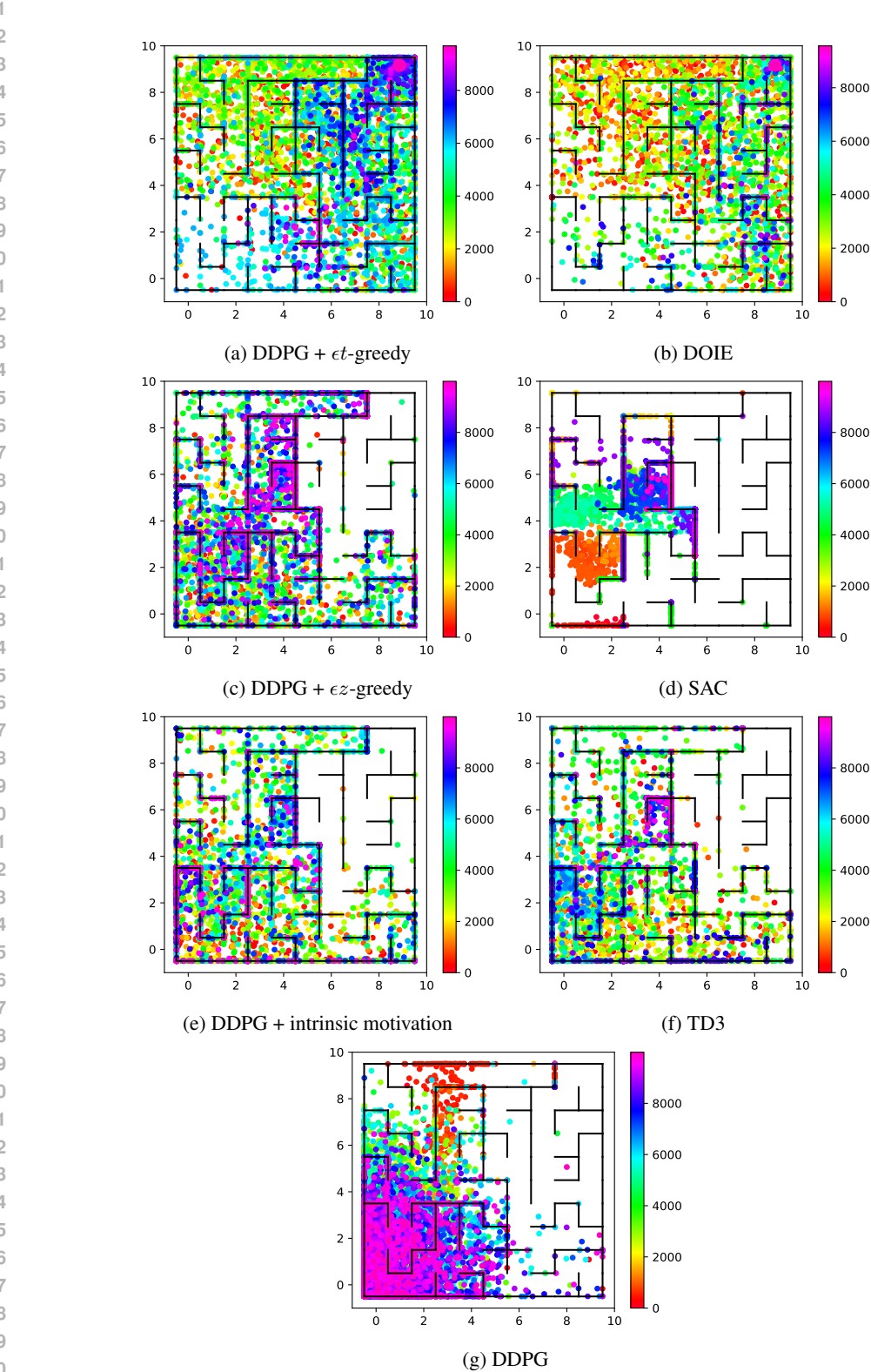

Figure 7: The agent's location at the end of episodes throughout the training in Wall-maze.

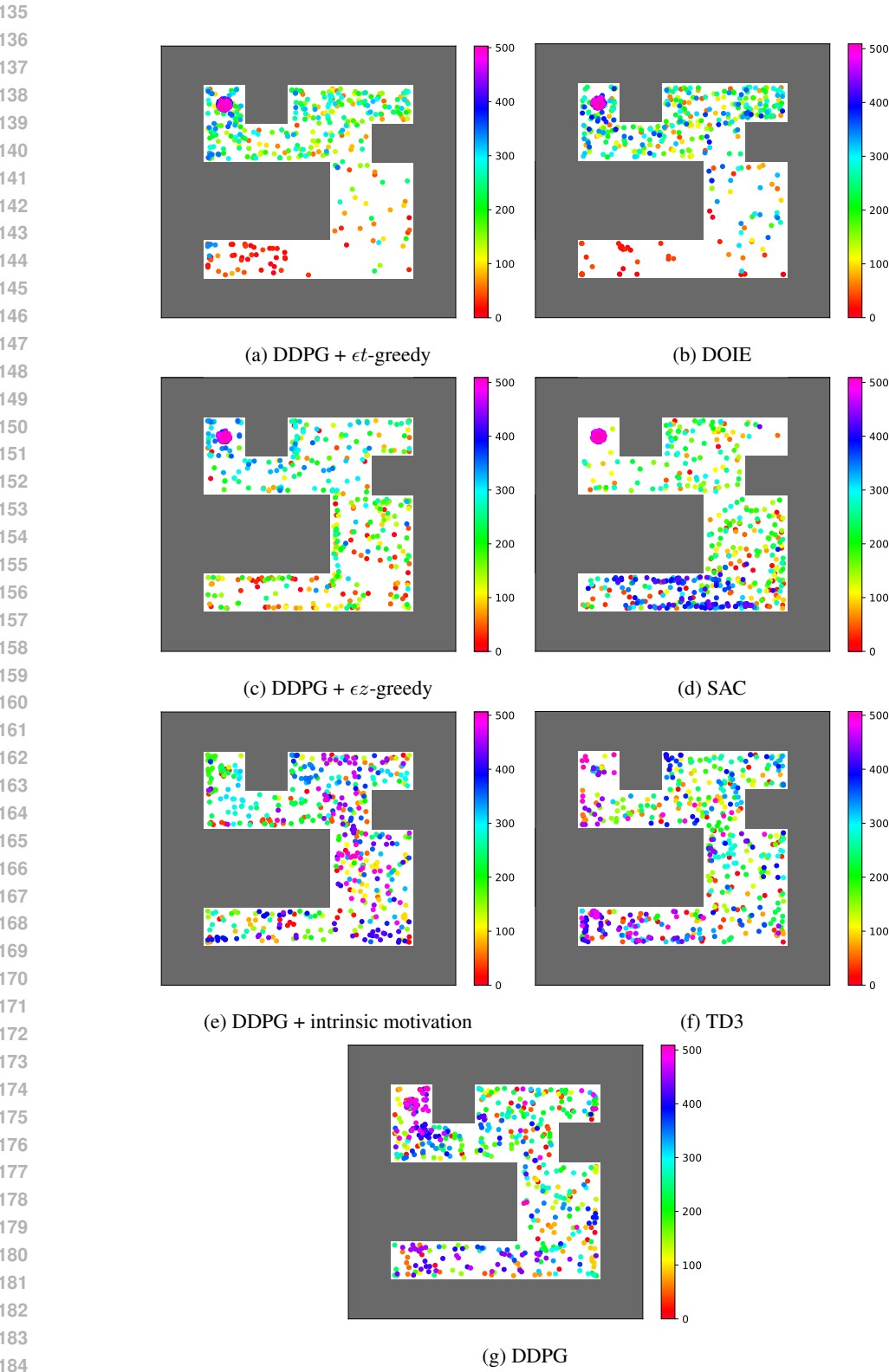

Figure 8: The agent's location at the end of episodes throughout the training in U-maze.

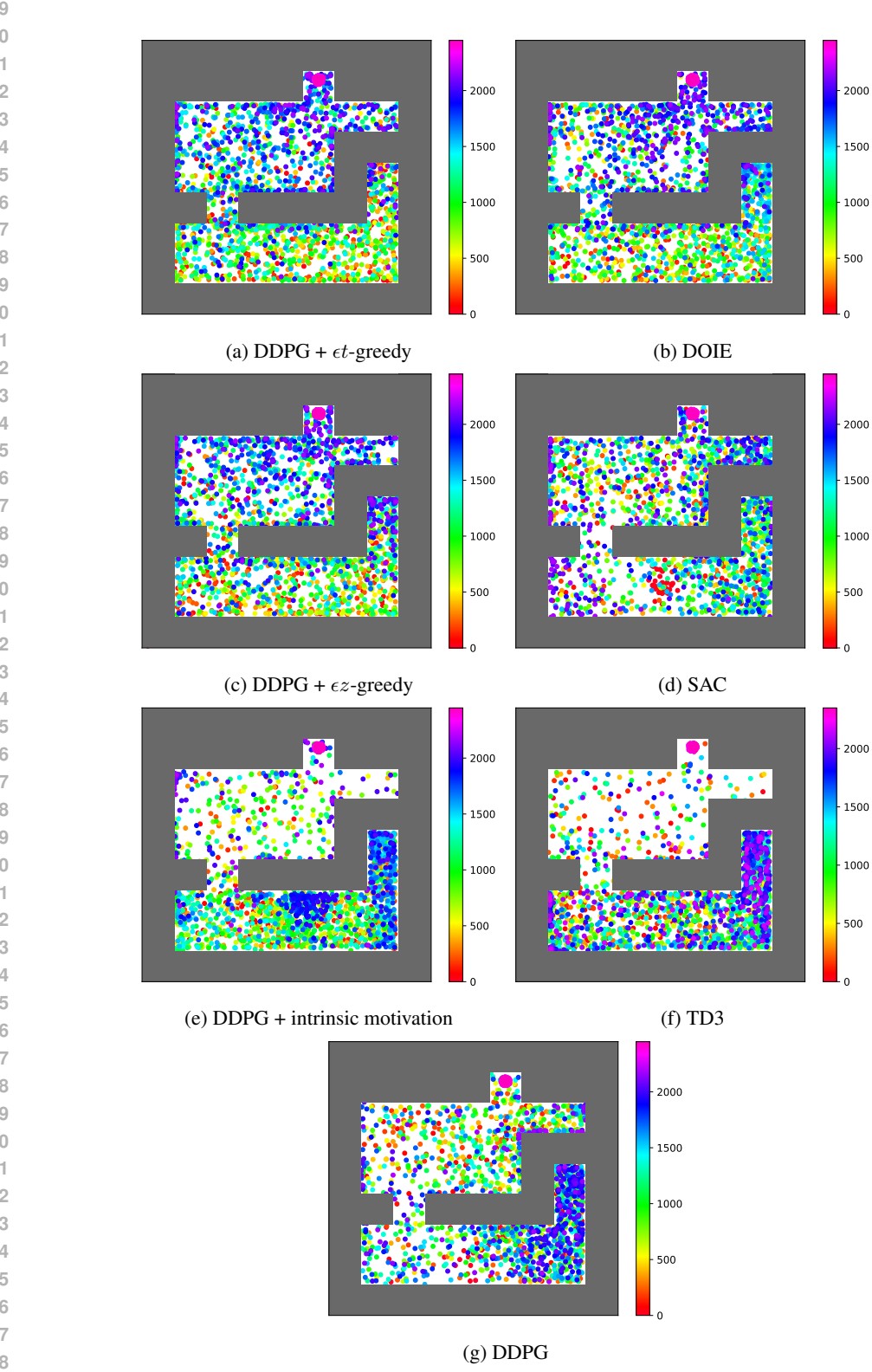

Figure 9: The agent's location at the end of episodes throughout the training in Point-push.

