# OpenReview forum: "ETGL-DDPG: A Deep Deterministic Policy Gradient Algorithm for Sparse Reward Continuous Control"
_ICLR.cc/2025/Conference — ICLR 2025 Conference Withdrawn Submission_

### Official Review · Reviewer_vfRg · 2024-11-01

**Soundness:** 2
**Presentation:** 3
**Contribution:** 2
**Rating:** 3
**Confidence:** 3

**Summary:**

This work combines three algorithmic ideas to improve the performance of DDPG in sparse-reward tasks. Fist, the authors propose $\epsilon$t-greedy exploration, which consists in building a graph-based representation of the environment, and navigating to nodes with a low visitation count. This method is accompanied by a formal analysis. Additionally, a buffer sampling technique is used to prioritize transitions with non-zero rewards, and longest n-step returns help quickly propagating values across trajectories. The resulting method, named ETGL-DDPG, outperforms simple algorithms in six continuous control tasks, demonstrating that each of the three components positively affects performance.

**Strengths:**

- The paper is well written and easy to follow. Contributions are outlined clearly and supported by experimental evidence.
- The limitations of hashing are acknowledged in the final discussion.

**Weaknesses:**

- Related works and missing baselines: this works does not sufficiently mention established methods for exploration in sparse reward settings. To the best of my knowledge, the standard method for sparse-reward RL is HER [1], which is only quickly mentioned in Section 5. Explaining why HER is not a relevant baseline would be very important. Otherwise, a comparison to it is expected. Moreover, the core exploration technique relies on a graph-search procedure. Similar ideas have been proposed in the past, both in the context of exploration [2, 3] and general value estimation [4]. Methods involving graph-search are very powerful, but require non-trivial implementation efforts and introduce complexity. None of the baselines considered relies on a graph; I would encourage the authors to clearly argue why the proposed graph search procedure is more appropriate than existing ones, or show how it outperforms them. More in general, the selection of baselines includes simple methods designed for the general RL problem, and not specialized to sparse-rewards.
- Novelty and orthogonality: the second and third technique (the dual replay buffer and longest n-step rewards) are not entirely novel, as acknowledged in lines 273 and 298. They appear to be applied without further analysis or changes. Moreover, the graph-based technique seems to work well $independently$ from the other two. It thus seems that the three proposed techniques are rather orthogonal. Why is it important to combine all three techniques? How do they relate to each other? In the current state, I do not appreciate the novelty in the second and third techniques, and thus wonder why the authors would not focus on the first one.

**Questions:**

Following the previous paragraph,
- Can the author provide a comparison to HER?
- Why is the proposed graph search procedure is more appropriate than existing ones? Does it outperform them?
- Are the three proposed techniques orthogonal?

Moreover, some minor questions and comments:
- Why is DDPG chosen as the base deep reinforcement learning algorithm, instead of relatively more modern algorithms such as TD3 [5] or SAC [6]?
- line 83: authors state that "the agent must achieve the goal many times to make sure that the reward is eventually propagated backward to early states". I am not coninced this statement holds. To the best of my knowledge, rewards are propagated when optimizing the TD loss for a given batch; therefore, it is sufficient to sample sufficient batches as long as the goal is achieved a single time in the buffer. I would ask the authors to comment on this.
- n-step returns are on-policy. It could be helpful to acknowledge this.
- line 88: typo (new temporally version) to (new temporally extended version)
- line 206: typo ($\delta$-optimal) to ($\epsilon$-optimal)
- line 246: typo, I believe ($\mathcal{P_W}$) to ($\mathcal{P_X}$)
- line 315: the chosen baselines are arguably not state-of-the-art for sparse-reward environments, see comments above.

**References:**

[1] Andrychowicz et al., Hindsight Experience Replay, NIPS 2017

[2] Ecoffet et al., First return, then explore, Nature 2021

[3] Gallouedec et al., Cell-Free Latent Go-Explore, arXiv 2023

[4] Eysenbach et al., Bridging Planning and Reinforcement Learning, NeurIPS 2019

[5] Fujimoto et al., Addressing Function Approximation Error in Actor-Critic Methods, ICML 2018

[6] Haarnoja et al., Soft Actor-Critic: Off-Policy Maximum Entropy Deep Reinforcement Learning with a Stochastic Actor, ICML 2018

---

### Official Review · Reviewer_ZA8A · 2024-11-02

**Soundness:** 3
**Presentation:** 3
**Contribution:** 2
**Rating:** 5
**Confidence:** 4

**Summary:**

This paper introduces a recipe of additions to DDPG that would make it particularly suitable for addressing sparse-reward (in particular, those with informative rewards only on the last transition of successful episodes). The recipe involves three main components: (1) $\epsilon t$-greedy exploration, (2) an additional episodic buffer only containing successful episodes, and (3) Monte Carlo updates on successful episodes instead of 1-step TD.

The most involved component of the approach is (1), which involves tree-search for finding a good option to execute. This process requires a model of environment's transition dynamics in principle, which the authors circumvent by using the replay buffer as a model and utilizing a SimHash model on a discretized variant of the 2D or 3D environments.

They evaluate the performance of this approach on several sparse reward testbeds: physics based 2D point-mass and 3D robotic manipulation tasks, as well as non-physical maze problems. The results show full coverage of discretized 2D or 3D spaces due to addition (1). Also, reaching success rates close to 1 on all tasks (on average), and generally well above other baselines.

**Strengths:**

- The proposed recipe will be useful for those interested in similar problem scenarios, e.g. using DDPG for tackling physical control problems.

- Paper is generally well-written, and ideas are mostly easy to grasp.

**Weaknesses:**

- Several related works are not sufficiently discussed in the context of the contributions:
    - Count-based methods in DRL: e.g. Ostrovski et al. (2017) is referenced but not discussed to any extent in connection to the SimHash method.
    - Hindsight Experience Replay is a relevant line of work, as it's addressing the same type of problems while remaining simpler and broader in applicability. While the paper is referenced, it is not used as a baseline nor discussed to an appropriate extent.

- Baselines are unfortunately somewhat weak; e.g. D3PG (non-distributed variant of D4PG) would be an important baseline to build on and/or to compare against. Specifically, DDPG + PER (as in D3PG) would have allowed to better assess the impact of the dual-memory approach. DDPG + N-step return (as in D3PG) would have allowed to better assess the difference between with a standard approach beyond 1-step TD updates. I wouldn't even worry about the distributional component of D3PG/D4PG, but combinations with PER and N-step returns are quite important to test against.

**Questions:**

1. Default exploration in DDPG is based on the OU noise, which is a temporally-correlated noise. As such, I'm curious if the "DDPG" baseline in, e.g., Fig. 5 is based on $\epsilon$-greedy or the standard OU noise? (And does your answer hold for all results for DDPG in the paper?)

2. Have you performed any leave-1-out ablations as well? I.e. similar to Fig. 5 but with the full algorithm minus one addition.

3. Do you have any comment on including DDPG + PER + N-step returns as a baseline?

4. Regarding the options generated by your tree-search approach: Will they remain valid in stochastic domains?

5. Why is there no Importance Sampling correction (similar to PER) to reduce the bias of sampling from successful episodes more frequently?

6. Why are there no experiments / detailed discussions around Hindsight Experience Replay?

7. Could you comment on connection of your approach wrt. count-based exploration techniques in DRL?

8. Is the discretization of "states" really discretizing the true state of the problem or just the 2/3 spatial locations?


### Minor comments:
L252: Comma should move to the end of equation.

---

### Official Review · Reviewer_UZxt · 2024-11-03

**Soundness:** 2
**Presentation:** 3
**Contribution:** 2
**Rating:** 3
**Confidence:** 4

**Summary:**

This paper presents three components that can be added to deep deterministic policy gradients (DDPG) algorithm to improve its performance. These components are (a) a new exploration strategy $\epsilon t$-greedy that performs exploration by building a tree to find the states at the frontier of its exploration and a path to get to those states, (b) a divided replay buffer to keep special track of successful episodes, and (c) changing the critic update to use T-step updates, turning the successful episode targets into Monte Carlo returns. The paper evaluates these techniques on three navigation and three manipulation tasks that require continuous-space actions, and show that the proposed method, ETGL-DDPG, outperforms some of the baselines compared to.

**Strengths:**

* This paper shows strong empirical results. The effect of the different components is clear from the experiments.
* The explanation of the three components is also fairly clear. Figure 1 was helpful in understanding the proposed techniques
* Some of the analysis figures are also well done. I particularly like Section A.5, which shows how the terminal state distribution for the different algorithms evolve over training.
* Training details are also extensive, and I believe the results in this paper could be reproduced
* The presented exploration strategy $\epsilon t$-greedy, is a combination of good ideas, and shows great promise.
* The sample complexity analysis adds theoretical depth to the paper.

**Weaknesses:**

* While the results in the paper appear strong empirically, there is some worry that methods that seem very similar to those proposed in the paper are not being compared to.
* The proposed $\epsilon t$-greedy exploration looks a lot like Go-Explore [1]. While there are certain differences (no behavioral cloning of the exploration policy), the similarities are close enough that they should be addressed in the paper or the method should be compared to. Additionally, the LSH method is very similar to #-exploration [2]. While perhaps comparing to the method itself might not be relevant since the results might not be state-of-the-art any longer, and the paper itself is cited, more detailed comparison to this technique in the related work would add depth to the paper.
* The use of two replay buffers, with one buffer used to store data leading to successes, sounds very much like [3]. Apart from the different replacement schemes (reservoir sampling vs FIFO), they seem like very similar ideas. Attribution and comparison (if relevant) would be good to have here.
* The longest n-step return is basically Monte Carlo updates for successes and very long N-step returns for failures. The change makes it (a) not use bootstrapping for successes, and (b) not bootstrap from the agent's current Q-values efficiently.  Could the authors justify this change would be better? Specifically, why the high variance updates from successes are preferred to bootstrapping, and why the very long n-step returns are preferred for failures? The ablations (Figure 5) seem to bear out that these long returns are actually not helping.
* Overall I feel like the exploration component is the main and helpful part of the paper. I would prefer the paper focus on this idea, and bring the analysis, such as Section A.5 into the main paper. That would be a much more impactful contribution, in my opinion.

Some minor nitpicks:
* Line 462: "except for soccer, where DDPG alone outperforms all baselines." The results show that $\epsilon t$-greedy outperforms DDPG there.
* Algorithm 1, line 24: typo: UnifromRandom($\phi(s_x)$)
* Algorithm 1, line 9: frontier nodes not being passed.
* Algorithm 1, line 10, counting function $n$ is not specified as an input, nor initialized.

References:

[1] Ecoffet, A., Huizinga, J., Lehman, J., Stanley, K.O. and Clune, J., 2021. First return, then explore. Nature, 590(7847), pp.580-586.

[2] Tang, H., Houthooft, R., Foote, D., Stooke, A., Xi Chen, O., Duan, Y., Schulman, J., DeTurck, F. and Abbeel, P., 2017. # exploration: A study of count-based exploration for deep reinforcement learning. Advances in neural information processing systems, 30.

[3] Kompella, V.R., Walsh, T., Barrett, S., Wurman, P.R. and Stone, P., Event Tables for Efficient Experience Replay. Transactions on Machine Learning Research.

**Questions:**

* What does the shaded portion in Figures 3 and 5 signify? Perhaps add it to the caption of the figure.

Other questions have been asked as part of the critique and feedback in previous section. Please refer to those.

---

### Official Review · Reviewer_Vg8Q · 2024-11-03

**Soundness:** 3
**Presentation:** 2
**Contribution:** 3
**Rating:** 5
**Confidence:** 2

**Summary:**

This paper addresses the limitations of the DDPG algorithm in sparse-reward environments. It identifies three main deficiencies: lack of directional exploration, uniform treatment of rewards in the replay buffer, and slow information propagation during policy updates. To tackle these issues, the authors propose three enhancements to DDPG, including $\epsilon t$-greedy, goal-conditioned dual replay buffer (GDRB), and longest n-step return. Empirical results demonstrate that each individual strategy enhances DDPG’s performance.

**Strengths:**

1. This work aims to enhance the most fundamental algorithms in RL, which I believe is very important.
2. The paper points out the important issue that DDPG is not suitable for sparse rewards settings and specifically identifies three drawbacks of DDPG.
3. The three improvements proposed in ETGL-DDPG sound effective to me in the sparse reward setting.
4. The benchmark experiments and ablation studies demonstrate the effectiveness of ETGL-DDPG.

**Weaknesses:**

1. This study proposes three improvements to DDPG in the context of sparse rewards, but I am uncertain about the degree of originality of each of these improvements. There is a substantial amount of related work on this topic (sparse reward, exploration, improving DDPG), and there are likely several studies closely related to each of the proposed enhancements, as some of them are straightforward and may be easy to conceive. Thus, a thorough literature review would be necessary to determine the novelty of the improvements. If these improvements (or some of them) have suffient innovation, I believe they would be valuable contributions.

**Questions:**

Please see weakness.

---

### Note · Authors · 2024-11-15

I have read and agree with the venue's withdrawal policy on behalf of myself and my co-authors.